# Inflammatory Profile Associated with Secondary Infection from *Bothrops atrox* Snakebites in the Brazilian Amazon

**DOI:** 10.3390/toxins15090524

**Published:** 2023-08-26

**Authors:** Távila Tatiane Amorim Cavalcante, Márcia Beatriz Silva de Souza, Juliana Costa Ferreira Neves, Hiochelson Najibe Santos Ibiapina, Fabiane Bianca Albuquerque Barbosa, Karolaine Oliveira Bentes, Eliane Campos Alves, Hedylamar Oliveira Marques, Monica Colombini, Suely Vilela Sampaio, Manuela Berto Pucca, Iran Mendonça da Silva, Luiz Carlos de Lima Ferreira, Vanderson de Souza Sampaio, Ana Maria Moura-da-Silva, Allyson Guimarães Costa, Wuelton Marcelo Monteiro, Jacqueline Almeida Gonçalves Sachett, Marco Aurélio Sartim

**Affiliations:** 1Programa de Pós-Graduação em Imunologia Básica e Aplicada, Instituto de Ciências Biológicas, Universidade Federal do Amazonas-UFAM, Manaus 69080-900, Brazil; 2Departamento de Biologia, Universidade Federal do Amazonas-UFAM, Manaus 69080-900, Brazil; 3Programa de Pós-Graduação em Medicina Tropical, Universidade do Estado do Amazonas-UEA, Manaus 69040-000, Braziljac.sachett@gmail.com (J.A.G.S.); 4Instituto Federal de Educação, Ciência e Tecnologia do Amazonas-IFAM, Manaus 69025-010, Brazil; 5Diretoria de Ensino e Pesquisa, Fundação Hospitalar de Hematologia e Hemoterapia do Amazonas-HEMOAM, Manaus 69050-001, Brazil; 6Laboratório de Imunopatologia, Instituto Butantan, São Paulo 05503-900, Brazil; 7Faculdade de Ciências Farmacêuticas de Ribeirão Preto, Universidade de São Paulo-USP, Ribeirão Preto 14040-903, Brazil; 8Curso de Medicina, Universidade Federal de Roraima-UFRR, Boa Vista 69310-000, Brazil; 9Departamento de Análises Clínicas, Faculdade de Ciências Farmacêuticas, Universidaed Estadual Paulista-UNESP, Araraquara 14800-903, Brazil; 10Diretoria de Ensino e Pesquisa, Fundação de Medicina Tropical Dr. Heitor Vieira Dourado–FMT-HVD, Manaus 69040-000, Brazil; 11Pró-Reitoria de Pesquisa e Pós-Graduação, Universidade Nilton Lins, Manaus 69058-030, Brazil

**Keywords:** snakebites, *Bothrops*, secondary bacterial infection, inflammation, cytokines, chemokines

## Abstract

*Bothrops* snakebite envenomation (SBE) is consider an important health problem in Brazil, where *Bothrops atrox* is mainly responsible in the Brazilian Amazon. Local effects represent a relevant clinical issue, in which inflammatory signs and symptoms in the bite site represent a potential risk for short and long-term disabilities. Among local complications, secondary infections (SIs) are a common clinical finding during *Bothrops atrox* SBE and are described by the appearance of signs such as abscess, cellulitis or necrotizing fasciitis in the affected site. However, the influence of SI in the local events is still poorly understood. Therefore, the present study describes for the first time the impact of SBE wound infection on local manifestations and inflammatory response from patients of *Bothrops atrox* SBE in the Brazilian Amazon. This was an observational study carried out at the Fundação de Medicina Tropical Dr. Heitor Vieira Dourado, Manaus (Brazil), involving victims of *Bothrops* SBE. Clinical and laboratorial data were collected along with blood samples for the quantification of circulating cytokines and chemokines before antivenom administrations (T0) and 24 h (T1), 48 h (T2), 72 h (T3) and 7 days after (T4). From the 94 patients included in this study, 42 presented SI (44.7%) and 52 were without SI (NSI, 55.3%). Patients classified as moderate envenoming presented an increased risk of developing SI (OR = 2.69; CI 95% = 1.08–6.66, *p* = 0.033), while patients with bites in hands showed a lower risk (OR = 0.20; CI 95% = 0.04–0.96, *p* = 0.045). During follow-up, SI patients presented a worsening of local temperature along with a sustained profile of edema and pain, while NSI patients showed a tendency to restore and were highlighted in patients where SI was diagnosed at T2. As for laboratorial parameters, leukocytes, erythrocyte sedimentation ratio, fibrinogen and C-reactive protein were found increased in patients with SI and more frequently in patients diagnosed with SI at T3. Higher levels of circulating IL-2, IL-10, IL-6, TNF, INF-γ and CXCL-10 were observed in SI patients along with marked correlations between these mediators and IL-4 and IL-17, showing a plurality in the profile with a mix of Th1/Th2/Th17 response. The present study reports for the first time the synergistic effects of local infection and envenoming on the inflammatory response represented by local manifestations, which reflected on laboratorial parameters and inflammatory mediators and thus help improve the clinical management of SI associated to *Bothrops* SBE.

## 1. Introduction

Snakebite envenomation (SBE) represents a relevant public health issue among tropical countries. Biologically active toxins from snake venoms are responsible for the disturbance of several physiological events including hematoxic, neurotoxic, myotoxic, and target organs disfunctions, promoting local and systemic damage to patients [1,2]. The role of venom components and its mechanisms involved in envenomation pathophysiology comprise the major efforts in the SBE research; however, other clinical complications aside from the direct action of toxins, such as secondary infections (SI), are also associated with SBE morbidity [3,4].

*Bothrops* genus is the major perpetrating group of snakes causing SBE in Brazil, and *Bothrops atrox* (*B. atrox*) is responsible for most cases in the Amazon region [5,6,7,8]. The envenoming caused by this species is evidenced by several local manifestations comprised by pain, edema, increased local temperature and erythema, and it may evolve to severe complications related to tissue injury with possible long-term disabilities [9,10]. The inflammatory response secondary to the proteolytic effect of the toxins, especially metalloproteases, is the major pathophysiological mechanism involved in local events, which is associated to the recognition of the toxins by innate immunity and the releasing of damage-associated molecular patterns (DAMPs) [9,10,11,12,13]. However, another indirect mechanism involves secondary bacterial infections at the bitten site [14].

SI are important local complications from SBE, and its diagnosis is described by the appearance of signs as abscess, cellulitis or necrotizing fasciitis, along with other manifestations also attributed to venom activity such as erythema, hotness, swelling, and/or pain [14,15,16]. The incidence of this event varies by the aggressor species and geographical localization, with frequencies up to 77% in some sites [17,18]. Among the bacterial species, it has been reported that both Gram-positive and Gram-negative bacteria, as well as anaerobic, were associated with this complication; infection is probably caused by the inoculation of bacteria from the mouth, fangs, or venom microbiota [3,15,17]. The SI treatment consists of the empiric use of antibiotics, such as amoxicillin/clavulanate or ampicillin/sulbactam, and ciprofloxacin; however, resistance to this empiric strategy has been reported [15,17,19]. Therefore, the treatment of wound infection is pivotal to avoid extensive local tissue damage and sepsis [2,16].

Previous studies have shown that SI is an important complication resulting from *B. atrox* SBE in the Brazilian Amazon, accounting for about 40% of patients. Among clinical and laboratorial evaluation, SI was associated to increased levels of fibrinogen, alanine transaminase and C-reactive protein, intense pain, and moderate SBE severity [14]. Moreover, clindamycin was the most prescribed antibiotic as the primary antibiotic regimen; however, in 24.4% of the SI patients, this antibiotic regimen fails [20]. Although the relevance of the inflammatory reaction on local effects during *Bothrops* SBE is clinically recognized, studies on the repercussion of SI on the inflammatory response is still neglected. The present study describes for the first time the inflammatory profile associated with SI following *Bothrops* SBE.

## 2. Results

From the 94 patients included in this study, 42 patients presented SI (44.7%) and 52 did not present SI (55.3%) during the follow-up (Figure 1). From the 42 SI patients, 6 (14.3%) were diagnosed at T1, 15 (35.7%) were diagnosed at T2, 8 (19.0%) at T3, and 13 (31%) patients were diagnosed from T3 to T4. No patients were diagnosed with sepsis during the follow-up.

### 2.1. Factors Associated to Secondary Infections

The epidemiological characterization showed a predominance of SBE in male patients (85.1%), mainly from rural areas (92.5%), with the age group predominantly between 21 and 40 years (39.8%), and most were work-related accidents (47.3%). Although the most frequent affected anatomical site was the lower limbs, specifically the feet (62.4%), bites in hands showed a protection factor to develop SI with a 80% reduced risk compared to foot (OR = 0.20; CI 95% = 0.04–0.96, *p* = 0.045). The time elapsed from the bite to the medical assistance was up to 3 h in 61.7% of the cases, and 14.0% of the participants reported a previous history of SBE. In addition, 37.6% used traditional topical medications, 33.3% used oral medications, and 8.5% reported comorbidities. Tourniquet was used in 23.7% of cases. Interestingly, patients classified as moderate were over two times more likely to develop SI (OR = 2.69; CI 95% = 1.08–6.66, *p* = 0.033) (Table 1).

### 2.2. Local Clinical Manifestations Analysis

As observed in Figure 2A, both SI and NSI groups showed a sustained edema (edema ratio above 1) from T0 to T4. Although the SI group presented slightly higher values, there was no significant difference between groups. Regarding local temperature, the SI group showed an increasing pattern in local temperature over time, which was significantly higher at T3 and T4 (Figure 2B). In terms of pain intensity, an overall analysis shows a pain decrease intensity during follow-up in both groups (Figure 2C). However, the frequency of patients with intense/moderate pain was higher in the SI group compared to the NSI group at T1, T2 and T4 (Figure 2C).

Although some differences in local manifestations could be observed among all patients regarding SI vs. NSI, the onset of secondary infections was recorded at different days during clinical evolution. Therefore, to evaluate patients from a particular perspective, the SI group was segmented according to the time at which infection was diagnosed (T1 to T3), and the local parameters profile was analyzed in every time. Considering that the T4 group comprises patients with SI diagnosis in different times between T3 and T4, the T4 group was not analyzed. As observed in Figure 3, edema at T3 and T4 was greater in patients with SI diagnosed at T2 compared to the NSI group, which is in parallel with an increase in edema in the SI group and decrease in the NSI group. As for temperature, we also observed higher values at T1, T2 and T4 in patients with SI diagnosed at T2 compared to NSI, which was accompanied by an increment at the times mentioned. The frequency of intense/moderate pain was also found to be higher at T1, T2 and T3 in patients with SI diagnosed at T2. No statistical differences were observed in patients with diagnosis at T1 and T3 in the NSI group.

### 2.3. Laboratory Parameters

Along with clinical manifestations, hematological and biochemical laboratory parameters were investigated (Figure 4). Blood leukocytes showed peak values at T0 and decreasing over time for both SI and NSI patients. Among the patient groups, SI individuals presented higher counting at T0 and T3 compared to NSI. As for neutrophils, a similar pattern was observed during the follow-up, with the SI group presenting increased values at T2 and T3. Erythrocyte sedimentation ratio (ESR) values exhibited a rising profile over the days, peaking at T4 for both groups, with differences between SI and NSI at T3 and T4. The frequency of patients with values of C-reactive protein (CRP) above 6.5 mg/dL was higher in the SI group from admission (T0) and during all follow-up (T1 to T4). For both SI and NSI groups, fibrinogen levels showed a rising during clinical evolution, with superior levels in SI from T2 to T4. No significant differences among SI and NSI were found for creatine kinase (CK), lactate dehydrogenase (LDH), and alanine aminotransferase (ALT) (Figure 4).

When the SI group was segmented according to the day of infection diagnosis (Figure 5), we observed that leukocytes were found higher at T0 in patients with diagnosis of T1. As for neutrophils, the difference was noticed during all days of follow-up (T0–T4) for patients with infection confirmed at T3. In a similar manner, patients diagnosed with SI at T3 presented increased values of ESR from T0 to T4. Concerning CRP, we observed that the frequency of patients with elevated values was higher at T3 and T4 when infection confirmation was at T2, and as for patients with SI confirmation at T3, the difference was found from T2 to T4. No significant changes were found for CK, LDH, fibrinogen and ALT (Figure 5).

### 2.4. Soluble Immunological Molecules Profile

The detection of cytokines and chemokines in blood serum was also evaluated. As demonstrated in Figure 6, cytokine IL-6 was found increased from T0 to T3 in SI patients, showing a gradual decrease over clinical evolution. As for IL-10, levels at T0 were higher in the SI group as well as chemokine CXCL-8 at T1. No significant changes were found for IL-2, IL-4, IL-17, TNF-α, INF-γ, CCL-2, CCL-5, CXCL-9 and CXCL-10 (Figure 6).

The evaluation of cytokines profile during follow-up based on the diagnostic day (Figure 7) showed that patients with SI determined at T1 presented increased levels of IL-6 at T0 and T2, which was accompanied by IL-10 at T0. As for patients diagnosed at T2, they presented higher levels of IL-6 at T1. Diagnosis of SI at T3 was related to increased levels of IL-6 from T0 to T3. Others novel associations with SI were found, where patients with SI determined at T2 presented augmented levels of IL-4 and TNF-α at T0 compared to the NSI group. As for patients with SI determined at T3, higher levels of IL-2 at T1, INF-γ at T2, and TNF-α at T4 were observed. No significant changes were found for IL-17, CCL-2, CCL-5, CXCL-9 and CXCL-10 (Figure 7).

As for chemokines, patients with SI diagnosed at T1 presented increased levels of CXCL-8 at T0 compared to the NSI group. In patients with infection confirmed at T3, both CXCL-9 and CXCL-10 levels were higher from T0 to T4. No significant changes were found in patients diagnosed at T2 (Figure 8).

A network analysis of interactions between the immunological molecules, laboratorial and clinical parameters was performed and compared between overall SI patients, patients stratified by the day of SI diagnosis (SI segmented patients), and NSI patients. As illustrated in Figure 9, a high number of correlations between clinical, laboratorial and inflammatory mediators were observed from T2 onwards in each group of patients; several of them were commonly found in all three groups, and most were between cytokines and chemokines. However, SI overall and segmented groups presented an increased number of moderate and strong positive correlations compared to NSI patients, and most were also between cytokines and chemokines (Figure 9).

As concerning the distinct correlations among the three groups of patients, the SI overall group presented increased numbers of positive correlations predominantly involving cytokines and chemokines, highlighting interactions between IL-2, IL-10, IL-6, CCL-2, IL-17 and CXCL-10 to other cytokines and chemokines (Figure 9). In NSI patients, a higher number of correlations were observed between laboratorial parameters and cytokines/chemokines, among which ALT, VHS, LDH, neutrophil and leukocytes showed more prominent positive correlations to IL-10 and CXCL-10, and they were negatively correlated to CCL-5. Although less frequent, laboratorial parameters have also shown positive correlations with cytokines/chemokines in the SI overall group, highlighting CK and fibrinogen. As for the SI-segmented group, clinical and laboratorial parameters were found to present markedly positive correlations among them and with cytokines/chemokines, highlighting increased numbers of correlations involving VHS, fibrinogen, edema and CK to CXCL-8, to IL-4, IL-10 and TNF in patients diagnosed at T1 and T2. In patients diagnosed at T3, cytokines interactions between IL-2, IL-4, IL-10, and TNF are found more pronounced (Figure 9).

## 3. Discussion

Secondary infection (SI) following snakebites represents a relevant clinical issue during patient care. On one hand, local infection has distinct aspects from envenomation per se, which is mainly related to management and treatment involving antibiotic therapy [14]. However, aspects related to the local inflammatory process that result in clinical signs and symptoms may present a synergism in relation to the action of the venom. Thus, the present study, for the first time in the literature, evaluates the role of SI on the local effects and inflammatory profile of *Bothrops* SBE patients.

Regarding the population of the present study, male patients of active working age and residents of rural areas constituted the majority of the patients, where 45% evolved to SI against 40% previously described for *Bothrops* snakebite victims in the Brazilian Amazon [14]. As risk factors for developing SI, our results showed that patients classified as moderate envenoming were more vulnerable. The Brazilian Ministry of Heath guidelines standardize the classification based on local and systemic clinical manifestations. Local effects evaluation is more prominently used to differentiate mild and moderate cases, ranging from discrete local signs of pain and swelling (mild) to evident edema and ecchymosis without necrosis (moderate). However, severe cases are more distinguished by systemic manifestations as severe bleeding, cardiovascular impairment and/or acute renal failure [21]. Therefore, cases classified as moderate constitute patients whose local signs are markedly evident in terms of edema size and extension along with pain. Considering that the clinical effects of SI are often characterized by the classical signs and symptoms of inflammation [22], *Bothrops* envenomations (independently of SI appearance) are also characterized by manifestations such as pain, heat, swelling, and redness [9,23]. Another interesting result observed in the present study is that a lower risk for the development of SI was observed when the biting site was the hands compared to feet. This is possibly due to a better asepsis of hands, considering that the majority of patients are from rural areas, and accidents take place in agricultural land or on trails used to access workplaces or river margins, with a rare use of footwear; snakebites on low limbs may facilitate contamination and SI [24]. We have also observed that only six patients were diagnosed at 24 h and the majority of 15 patients were diagnosed at 48 h. Considering that the diagnosis of SI is based on clinical and laboratory parameters that are associated with the inflammatory process, and that envenomation also causes similar effects, we observed fewer cases of SI diagnosis at 24 h due to the overlapping of the inflammatory effects of infection and snakebite envenomation, making it more difficult to diagnose SI based on clinical/laboratory results. However, from 48 h, the effects of envenomation tend to reduce, mainly due to antivenom therapy, thus making it easier to distinguish clinical/laboratory alterations of SI from envenomation.

The Inflammatory response is the main mechanism triggering local effects during *Bothrops* SBEs, and it is responsible for edema, pain, increase local temperature, erythema, and possible functional loss [23]. Considering that SIs are also responsible for the same phlogistic signs, we have evaluated the local signs and symptoms—pain, edema and local temperature—during from SI and NSI patients. A global analysis showed that all three signs and symptoms were aggravated in SI patients. Also, the results indicate that the association of SI and venom effect is responsible for worsening local temperature during the follow-up, along with a sustained profile of edema and pain, while NSI patients shows a tendency to restore these parameters. Therefore, the present results clearly demonstrate a synergism of venom and infection-induced local inflammatory signs and symptoms by increasing the extension of the disturbance evaluated. We also analyzed the local parameters from the SI group based on the day of infection diagnosis in order to better assess the time course of the local clinical signs and symptoms. It was observed that patients with infection confirmation at T2 showed an increasing in edema on the day after diagnosis that sustained up to T4, a higher temperature profile starting the day before and sustained to T4, and an increased frequency of patients with intense/moderate pain from the day before diagnosis and sustained up to the day after. Thus, the results show that the synergistic effects of venom and infection on local signs and symptoms are more clearly distinguished in patients diagnosed with SI at T2. *Bothrops* snake venoms are known to induce inflammation by a direct mechanism of toxins to trigger immune cells (and soluble mediators) as well by releasing intrinsic molecules resulting from tissue damage [11,25,26,27]. Similarly, SI shares similar mechanisms to initiate inflammatory response, such as the direct activation of pathogen recognition receptors on leukocytes and the complement system as well as the generation of DAMPs [28,29]. Therefore, the increased and persistent inflammatory response observed in SI patients is responsible for a more persistent edema, pain and local temperature increase. Another interesting factor is that the edematogenic effect on NSI patients was sustained during the follow-up. These results indicate the limitation of antivenom therapy toward local effects, differently from others systemic events such as coagulopathy [30,31].

Laboratorial assays are key procedures during patients’ management in which different hematological and biochemical assays are performed to measure pathological disturbances [4]. The present study evaluated laboratorial markers involved in inflammation (total leukocyte and neutrophil count, CRP and ESR), liver alterations (ALT), tissue damage (CK and LDH) and inflammation/coagulation (fibrinogen) markers. Our study has shown that the total leukocytes, neutrophils, ESR, fibrinogen and CRP were found increased in patients with SI. When analyzed focusing on the day of diagnosis, patients with SI confirmation at T3 presented marked augmented values of neutrophils, ESR and CRP during most periods of follow-up. Interestingly, these markers were found to be increased in the patients days before the SI diagnosis, indicating them as possible predicting factors for SI at T3, including increased total leukocyte counting at T0 for patients with confirmation at T1. Therefore, the results confirm a synergistic inflammatory response of local infection and venom observed by the increment of the classical inflammatory markers levels as leukocyte and neutrophil counting, ESR, and CRP, confirming the above results on local clinical signs and symptoms.

Leukocytosis and neutrophilia are common findings in *Bothrops* SBE, and they have been previously described to be directly associated with SBE severity, SI and local complications as necrosis [14,32]. Therefore, increased levels of circulating leukocytes and neutrophils could work as predictor factors to wound infection and possible local complications. We have also observed that ESR was also elevated on a late phase (from T3 onwards) in patients with secondary infection. As for fibrinogen, although its levels were found increased in the overall patients with SI, no difference was observed in the analysis of patients diagnosed with SI at T1, T2 or T3, indicating that possibly the significant differences could be observed in the patients diagnosed after T3 but were not included in the analysis. Although fibrinogen is a hallmark molecule in hemostasis, inflammation leads to an increased hepatic expression and increased circulating of the protein, being considered as an acute inflammatory marker [33,34]. However, it well known that *Bothrops* envenomings are responsible for coagulation disturbances induced by hemostatically active toxins, which leads to fibrinogen consumption [35]. Therefore, in the present study, we observed that the fibrinogen levels are found lower in the beginning of follow-up due to venom-induced coagulopathy. However, the levels over time are increased and were found to be higher in patients with SI as a result of the inflammatory response. Therefore, fibrinogen may be considered a potential marker of SBE-related SIs specifically after coagulopathy restoring.

Inflammatory mediators perform an important role during *Bothrops*-induced inflammatory response, in which cytokines and chemokines are responsible for coordinating the immune response [36]. In the present study, we have evaluated the blood cytokines and chemokines profile and its correlation to laboratorial and clinical parameters, and we analyzed its association to SI. The cytokines IL-10, IL-6, TNF and CXCL-8 contrasted among mediators, which were increased during specific moments on SI patients along with IL-2, INF-γ, IL-4, CXCL-9 and CXCL-10 when evaluated from patients with SI segmented based on the diagnosis day. Similar findings from the previous study on *Bothrops* envenomation in the Brazilian Amazon showed that different cytokines/chemokines, including IL-2, IL-6, IL-10 and CXCL-8, were found increased in patients with severe tissue complications, correlating the plasma levels of these mediators to the local manifestations [25]. In our study, IL-6 was found increased compared to NSI patients during most of the follow-up evaluations. Pre-clinical studies using an in vivo approach of *Bothrops* envenoming have shown that IL-6 is a key mediator during the inflammatory response, in which neutrophils are a major source of production [37]. Interestingly, IL-6 and IL-10 were shown to be interesting predictor mediators for SI diagnosed from T1 to T3. As for wound infections, previous studies have shown that patients in medical assistance for SI present elevated levels of IL-6, lasting up to 5 days after intervention [38,39].

The kinetic profile and correlations of cytokines during the follow-up show that the SI patients present a marked combination of Th1/Th2/Th17 responses, with the predominance of the Th2/Th1 profile during the entire follow-up and a Th17 profile at T2 and T3. As for the NSI patients, the circulating response profile shows a less evident polarization to the Th1/Th2 response at T0 and following a Th1/Th17 pattern from T2. Ibiapina and colleagues [25] observed that a plurality of Th1/Th2/Th17 responses were observed following *Bothrops* envenomings, which were more evident in patients with severe tissue complications, corroborating with our findings and suggesting that SI is indeed associated to local complication. Concerning chemokines levels, CXCL-8, CXCL-9, and CXCL-10 were shown to be increased and associated with cytokines and other chemokines in SI patients, especially during the initial phase of envenoming. Previous studies have shown that wound infections present high chemokine levels, with a predominance of CXCL-8, CXCL-9, CXCL-10, and CCL-3, which impairs a rapid tissue healing [40].

Overall, SI patients showed an important positive correlations between cytokines/chemokines with fibrinogen levels from T3 onwards, indicating that the inflammatory response amplified by wound infection can increase fibrinogen production as described previously [41]. Meanwhile, NSI patients show a negative correlation at T0, indicating that the consumption of sera fibrinogen at this early stage is more associated to the venom direct activity on hemostasis and a possible association of the inflammation/coagulation crosstalk, as previously reported [42,43,44]. Association of the clinical parameters such as edema and local temperature with cytokines/chemokines are more evident in SI patients at T4, with focus of patients diagnosed at T2 and T3. As described above, the edema and local temperature in patients with SI were found sustained during follow-up up to T4, thus corroborating that a more complex inflammatory mechanism is associated with the clinical findings.

## 4. Conclusions

SBE-related SI is considered an important local complication and has been previously investigated mostly concerning the identification of the pathogenic agent and the proper antibiotic therapy. However, the present study has brought for the first time the evaluation of the inflammatory profile of wound infection and its implication on local clinical signs and symptoms, confirming that local infection and *Bothrops* venom-induced inflammatory effects act synergistically. The results show that patients with secondary infection diagnosis present worsened clinical manifestations concerning edema, pain and local temperature with a sustained impairment during follow-up evolution. The clinical findings were corroborated with classical hematological and biochemical laboratorial tests and associated with an inflammatory response profile. Increased neutrophils, ESR, CRP and total leukocyte values could be used as prognostic assays to predict the occurrence of infection. Moreover, the cytokine and chemokine patterns show a more evident mix of Th1 and Th2 response during the late phase of envenoming associated with the secondary infection inflammation, assigning some molecules, as IL-6 and IL-10, as possible novel markers for wound infection prediction. Therefore, the present study can help the improvement of clinical management of wound infections associated to *Bothrops* SBE in order to assist the development of future approaches to enhance the diagnosis and treatment of local infections.

## 5. Materials and Methods

### 5.1. Study Design, Participants and Ethical Aspects

This is an observational, longitudinal study carried out in adult patients who were victims of *Bothrops* SBE treated at the Fundação de Medicina Tropical Doutor Heitor Vieira Dourado (FMT-HVD) from July 2014 to July 2016. The FMT-HVD is a public reference center for the treatment of SBE in the city of Manaus, state of Amazonas, Brazil.

The study population consisted of a total of 94 patients diagnosed as victims of snakebite caused by *Bothrops* snakes, aged between 18 and 70 years, who had not undergone previous antivenom treatment before admission and without any sign of secondary infection at this time. The exclusion criteria consisted of pregnant women, children, and individuals either with self-reported immune or inflammatory diseases or under treatment with anti-inflammatory or immunosuppressive drugs. The clinical criteria for SBE management followed the protocol of the Ministry of Health of Brazil, and cases were classified as mild, moderate, or severe [21]. *B. atrox* SBE confirmation was performed by enzyme immunoassay [45]. All patients were treated with *Bothrops* antivenom according to the Brazilian Ministry of Health guideline [21].

The present study was approved by the Ethics Committee empirical Research by the FMT-HVD with CAAE: 19380913.6.3001.0005 (approval number 492.892/2014). All patients signed an informed consent form.

### 5.2. Admission and Follow-Up Procedures

Patients were followed up for 7 days, with clinical and laboratory evaluations being performed at 5 moments: at admission and before serum therapy (T0), 24 h (T1), 48 h (T2), 72 h (T3) and 7 days (T4) after admission. Approximately 4 mL of peripheral blood was collected by venipuncture at each time point in EDTA tubes (BD Vacutainer^®^ EDTA K2) or sodium citrate (BD Vacutainer^®^), and was plasma obtained, aliquoted and stored at −80 °C in order to measure chemokines and cytokines. Clinical–epidemiological data (gender, age, previous accident, zone of occurrence, SBE severity and affected anatomical region) and laboratory data were obtained from medical records from hospital routine.

### 5.3. Secondary Infection Diagnosis

SI was characterized by the observation of cellulitis and/or abscess at the bite site during hospitalization. Cellulitis was defined by the presence of local inflammation signs (erythema, edema, bruising and pain) with association to fever, leukocytosis, lymphangitis and/or lymphadenitis. An abscess was characterized by individual injuries, floating, presenting purulent secretion or serous–purulent secretion [14]. Patients who were diagnosed with SI after T3 (4, 5, 6 or 7 days after admission) were grouped as T4.

### 5.4. Clinical Parameters

#### 5.4.1. Edema

The edema of the local bite was evaluated by two different methods:

Circumference measurement ratio: a measuring tape (scale in centimeters) was used to measure the circumference of the affected limb near the bite site and on the contralateral limb. The results were expressed as percentage between the circumference ratio of the affected and contralateral limb.

Classification of edema: Edema was also evaluated by its extent in the affected limb, where it was classified as absent, mild (affecting 1–2 limb segments), moderate (affecting 3–4 limb segments) and severe (affecting more than 5 limb segments).

#### 5.4.2. Local Temperature

The measurement was performed as close to the bite site and in the identical anatomical region of the contralateral limb using an infrared digital clinical thermometer (Color Check AC322). The results were given as Δ temperature (temperature of the affected side—contralateral temperature).

#### 5.4.3. Pain

In order to assess pain intensity, a visual analogue scale was used, ranging from “no pain” to “pain as severe as it could be” with a value from 0 to 10, respectively, being classified into intense (rate 8 to 10), moderate (rate 4 to 7), mild (rate 1 to 3) and absent (rate 0), as described elsewhere [46].

### 5.5. Laboratory Parameters

Blood collected at each time point (T0, T1, T2, T3 and T4) was assessed for hematological and biochemical parameters, which was performed according to the routine of the Clinical Analysis Laboratory FMT-HVD. The methods consisted of leucocyte and neutrophil counting, creatine kinase (CK), lactate dehydrogenase (LDH), erythrocyte sedimentation rate (ESR), and C-reactive protein (CRP).

Soluble molecules CXCL-8, CCL-5, CXCL-9, CCL-2, CXCL-10, IL-2, IL-4, IL-6, IL-10, TNF, IFN-γ and IL-17A were quantified using the CBA (Cytometric Bead Array) technique. The BD™ Human Chemokine kits (BD^®^ Biosciences, Franklin Lakes, NJ, USA), and BD™ Human Th1, Th2, Th17 Cytokine (BD^®^ Biosciences, Franklin Lakes, NJ, USA) were used, following guidelines described by the manufacturer. For the acquisition of samples, the FACS Canto II Flow Cytometer (BD^®^ Biosciences, Franklin Lakes, NJ, USA) carried out an analysis performed using FCAP-Array™ software (v3.01).

### 5.6. Statistical Analysis

Statistical analysis of the data was performed using GraphPad Prism software (version 8.0.1). Initially, tests were performed to verify data normality using the Shapiro–Wilk test and demonstrated a non-parametric distribution. Therefore, data were normalized using log transformation, and a two-way ANOVA test followed by the Bonferroni post-test was performed for comparisons between SI vs. NSI groups and differences in time. Comparisons of frequencies of moderate/intense pain between SI vs. NSI were performed by the Chi-square test (corrected by Fisher’s test if necessary). Logistic regression was used for odds ratio estimation. For correlation analysis, the Spearman test was applied to the networks building, using the Cytoscape 3.0.3 software (Cytoscape Consortium, San Diego, CA, USA). Positive and negative correlations were considered significant when *p* < 0.05. The correlation pattern ® was used to categorize the strength of correlation as weak (r ≤ 0.35), moderate (r ≥ 0.36 to r ≤ 0.67) or strong (r ≥ 0.68). Statistical significance levels were considered as 0.05.

## Figures and Tables

**Figure 1 toxins-15-00524-f001:**
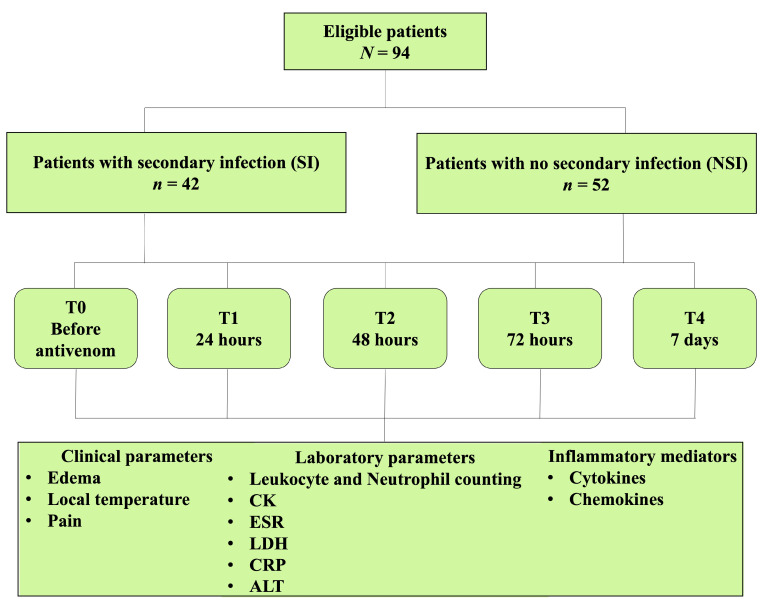
Study flow chart. Assessed and eligible patients categorized in patients with (SI) or without (NSI) secondary infection during the follow-up (T0–T4). Clinical, laboratory parameters and inflammatory mediators were evaluated for both groups at each time period. CK: creatine kinase; ESR: erythrocyte sedimentation rate, LDH: lactate dehydrogernase; CRP: C-reactive protein; ALT: alanine aminotransferase.

**Figure 2 toxins-15-00524-f002:**
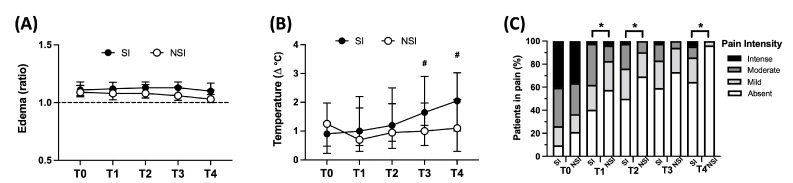
Local clinical parameters profiles between SI and NSI during follow-up. (**A**) Evaluation of edema (ratio affected limb vs. contralateral), (**B**) local temperature (∆°C affected limb vs. contralateral), and (**C**) frequency of individuals in different pain intensity (%) between patients with secondary infection (SI—black circle) and no secondary infection (NSI—white circle). (**A**,**B**) were reported as median values ± interquartile range. * *p* < 0.05 when comparing the values between SI vs. NSI at each time point (frequency between groups was compared considering moderated + intense), and ^#^ *p* < 0.05 when comparing differences in time within the same group for SI..

**Figure 3 toxins-15-00524-f003:**
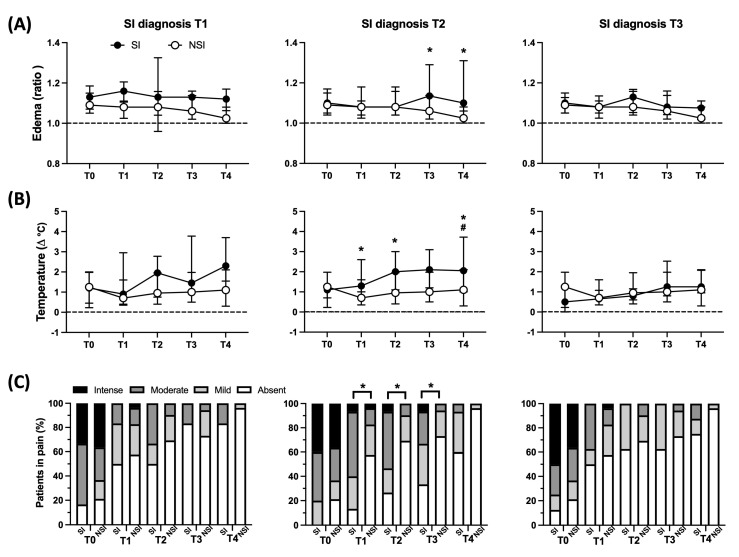
Local clinical parameters profiles between SI segmented on the day of diagnosis and NSI during follow-up. Patients with SI were segmented in different groups based on the day of diagnosis (T1, T2 or T3). (**A**) Evaluation of edema (ratio affected limb vs. contralateral), (**B**) temperature (∆°C affected limb vs. contralateral), and (**C**) frequency of individuals in different pain intensity (%) between patients with secondary infection (SI—black circle) and no secondary infection (NSI—white circle). (**A**,**B**) were reported as median values ± interquartile range. * *p* < 0.05 when comparing the values between SI vs. NSI at each time point (frequency between groups was compared considering moderated + intense), and ^#^ *p* < 0.05 when comparing differences in time within the same group for SI.

**Figure 4 toxins-15-00524-f004:**
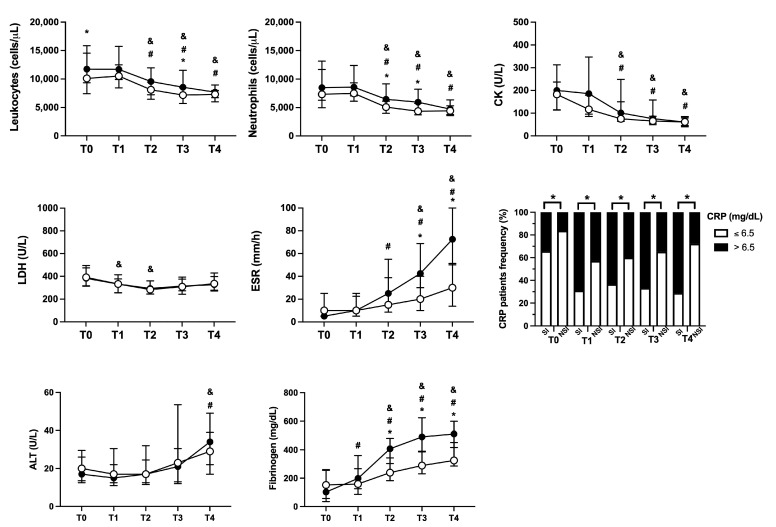
Laboratory parameters profiles between SI and NSI during follow-up. Evaluation between patients with secondary infection (SI—black circle) and no secondary infection (NSI—white circle) of total leukocytes, neutrophils, creatine kinase (CK), lactate dehydrogenase (LDH), erythrocyte sedimentation ratio (ESR), alanine aminotransferase (ALT), and fibrinogen were reported as median values ± interquartile range, while C-reactive protein (CRP) was reported as the frequency of individuals with values above (>6.5 mg/dL) or below (≤6.5 mg/dL). * *p* < 0.05 when comparing the values between SI vs. NSI at each time point; ^#^ *p* < 0.05 when comparing differences in time within the same group for SI, and ^&^ *p* < 0.05 when comparing differences in time within the same group for NSI.

**Figure 5 toxins-15-00524-f005:**
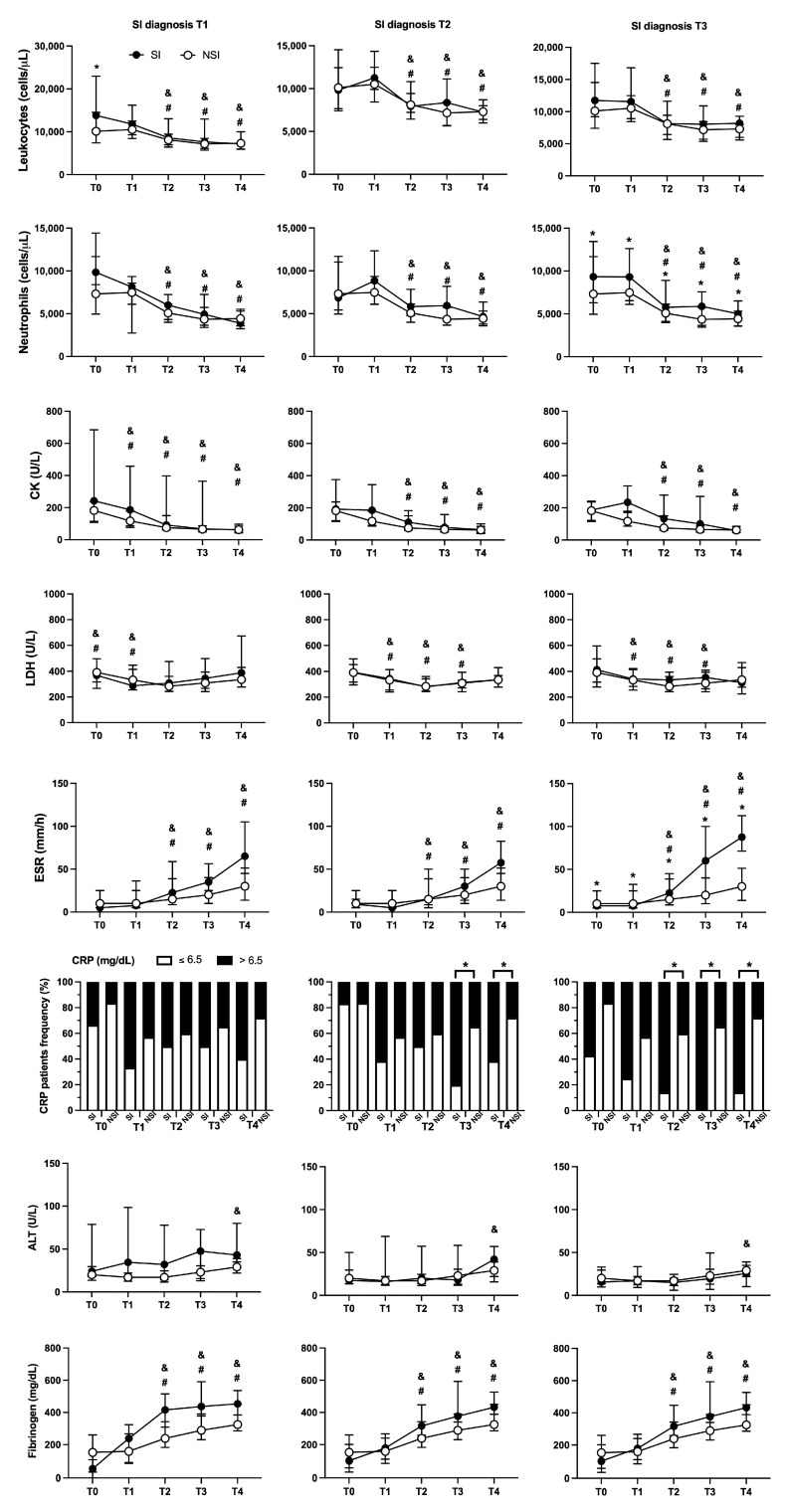
Laboratory parameters profiles between SI segmented on the day of diagnosis and NSI during follow-up. Patients with SI were segmented in different groups based on the day of diagnosis (T1, T2 or T3). Evaluation between patients with secondary infection (SI—black circle) and no secondary infection (NSI—white circle) of total leukocytes, neutrophils, creatine kinase (CK), lactate dehydrogenase (LDH), erythrocyte sedimentation ratio (ESR), alanine aminotransferase (ALT), and fibrinogen were reported as median values ± interquartile range, while C-reactive protein (CRP) was reported as the frequency of individuals with values above (>6.5 mg/dL) or below (≤6.5 mg/dL). * *p* < 0.05 when comparing the values between SI vs. NSI at each time point; ^#^ *p* < 0.05 when comparing differences in time within the same group for SI, and ^&^ *p* < 0.05 when comparing differences in time within the same group for NSI.

**Figure 6 toxins-15-00524-f006:**
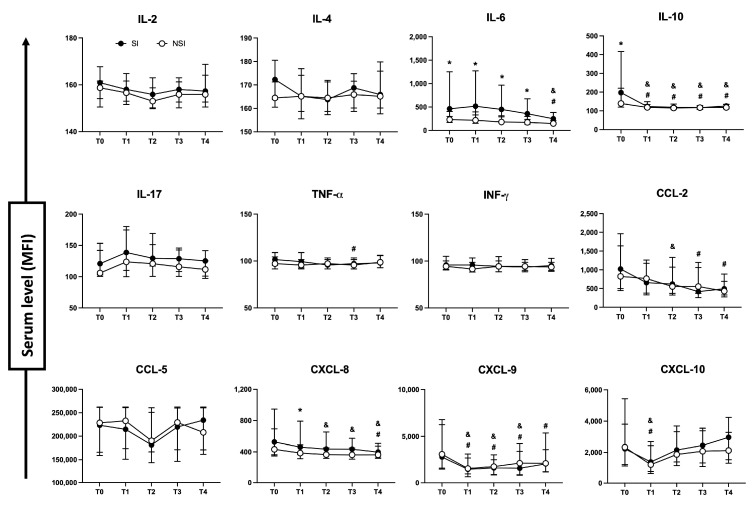
Soluble immunological molecules profiles between SI and NSI during follow-up. Evaluation between patients with secondary infection (SI—black circle) and no secondary infection (NSI—white circle) of cytokines and chemokines from blood serum, whose results are reported as median values ± interquartile range of median fluorescence intensity (MFI). * *p* < 0.05 when comparing the values between SI vs. NSI at each time point; ^#^ *p* < 0.05 when comparing differences in time within the same group for SI, and ^&^ *p* < 0.05 when comparing differences in time within the same group for NSI.

**Figure 7 toxins-15-00524-f007:**
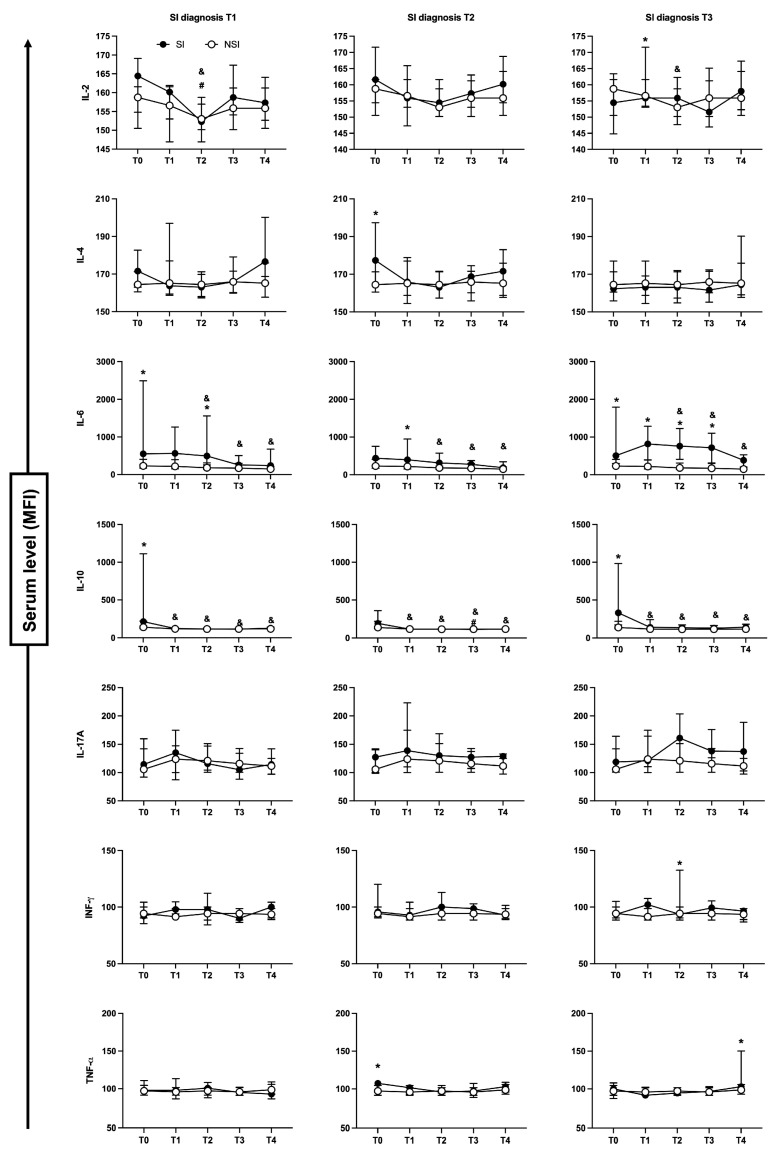
Cytokine profiles between SI segmented on the day of diagnosis and NSI during follow-up. Patients with SI were segmented in different groups based on the day of diagnosis (T1, T2 or T3). Evaluation between patients with secondary infection (SI—black circle) and no secondary infection (NSI—white circle) of cytokines from blood serum were reported as median values ± interquartile range. * *p* < 0.05 when comparing the values between SI vs. NSI at each time point; ^#^ *p* < 0.05 when comparing differences in time within the same group for SI, and ^&^ *p* < 0.05 when comparing differences in time within the same group for NSI.

**Figure 8 toxins-15-00524-f008:**
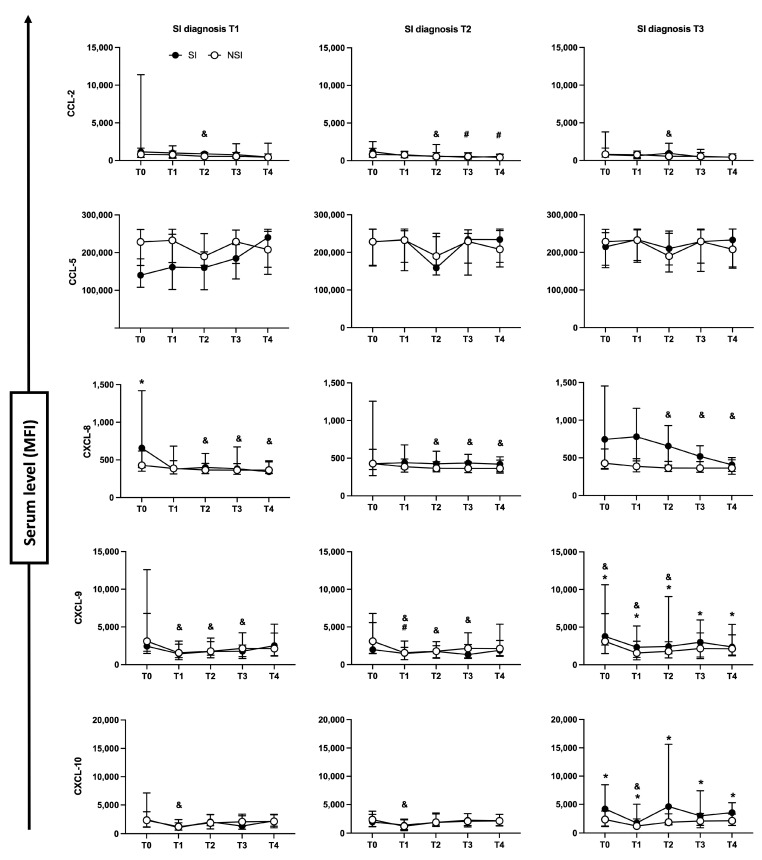
Chemokine profiles between SI segmented on the day of diagnosis and NSI during follow-up. Patients with SI were segmented in different groups based on the day of diagnosis (T1, T2 or T3). Evaluation between patients with secondary infection (SI—black circle) and no secondary infection (NSI—white circle) of chemokines from blood serum were reported as median values ± interquartile range. * *p* < 0.05 when comparing the values between SI vs. NSI at each time point; ^#^ *p* < 0.05 when comparing differences in time within the same group for SI, and ^&^ *p* < 0.05 when comparing differences in time within the same group for NSI.

**Figure 9 toxins-15-00524-f009:**
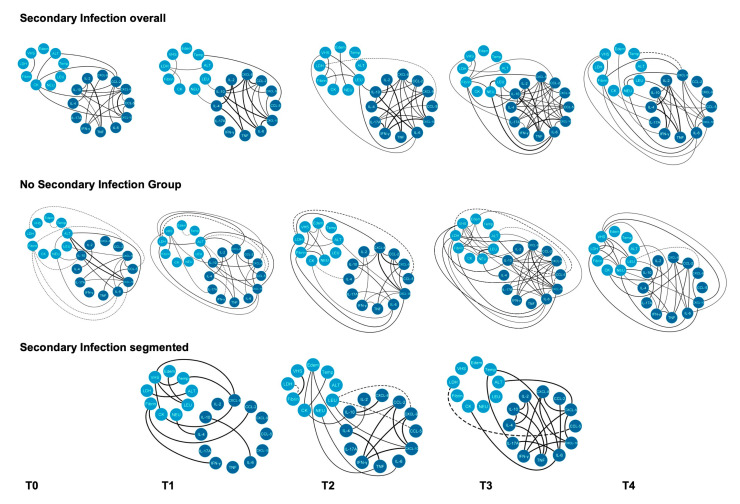
Network of soluble molecules, laboratory and clinical parameters interactions between groups in the follow-up. Each color group was used to identify the chemokines and cytokines (dark blue) and the laboratory and clinical parameters (light blue) between all secondary infection patients (SI overall), secondary patients segmented by the day of diagnosis (SI segmented) and patients without secondary infection (NSI). Dashed lines between molecules indicate a negative correlation while solid lines indicate a positive correlation. The thickness indicates the correlation strength. The correlation index (r) was used to categorize the strength of correlation into weak (r ≤ 0.35), moderate (r ≥ 0.36 to r ≤ 0.67), or strong (r ≥ 0.68).

**Table 1 toxins-15-00524-t001:** Clinical–epidemiological parameters from *Bothrops* snakebite patients.

Variables	Total (%) *N* = 94	SI (%) *n* = 42	NSI (%) *n* = 52	OR	CI 95%	*p* Value
**Gender**						
Female	14 (14.9)	6 (14.3)	8 (15.4)	0.92	0.29–2.89	0.882
Male	80 (85.1)	36 (85.7)	44 (84.6)	1	-	-
**Age (years)**						
0–10	5 (5.4)	4 (9.5)	1 (2.0)	1	-	-
11–20	11 (11.8)	5 (11.9)	6 (11.8)	0.21	0.02–2.52	0.217
21–40	36 (39.8)	16 (38.1)	21 (41.2)	0.19	0.02–1.87	0.155
41–60	27 (29.0)	13 (31.0)	14 (27.5)	0.23	0.02–2.36	0.217
>60	13 (14.0)	4 (9.5)	9 (17.7)	0.11	0.01–1.34	0.083
**Area of occurrence**						
Urban	7 (7.5)	3 (7.1)	4 (7.8)	1	-	-
Rural	86 (92.5)	39 (92.9)	47 (92.2)	1.11	0.23–5.24	0.899
**Bite site**						
Lower limbs	18 (19.4)	10 (23.8)	8 (15.7)	1	-	-
Upper limbs	2 (2.2)	0 (0.0)	2 (3.9)	-	-	-
Hand	15 (16.1)	3 (7.1)	12 (23.5)	0.20	0.04–0.96	0.045 *
Foot	58 (62.4)	29 (69.1)	29 (56.9)	0.80	0.28–2.32	0.681
**Work-related bite**	44 (47.3)	18 (42.9)	26 (51.0)	0.72	0.32–1.64	0.435
**Time to assistance (hours)**						
0–3	58 (61.7)	23 (54.7)	35 (67.3)	1	-	-
4–6	17 (17.0)	10 (23.8)	7 (13.4)	1.96	0.64–5.99	0.240
>6	19 (21.3)	9 (21.4)	10 (19.2)	1.52	0.55–4.23	0.421
**Previous snakebite history**	13 (14.0)	6 (14.3)	7 (13.7)	0.95	0.29–3.09	0.938
**Use of topical medicines**	35 (37.6)	14 (33.3)	21 (41.2)	0.71	0.31–1.67	0.438
**Use of oral medicines**	31 (33.3)	13 (31.0)	18 (35.3)	0.84	0.37–2.02	0.8261
**Use of tourniquet**	22 (23.7)	11 (26.2)	11 (21.6)	1.29	0.50–3.36	0.602
**Snakebite clinical classification**						
Mild	36 (38.7)	11 (26.2)	25 (49.0)	1	-	-
Moderate	48 (51.6)	26 (61.9)	22 (43.1)	2.69	1.08–6.66	0.033 *
Severe	9 (9.7)	5 (11.9)	4 (7.8)	2.84	0.64–12.65	0.171
**Antivenom vials administrated**						
2–4	15 (15.9)	4 (9.5)	11 (21.1)	1	-	-
5–8	59 (62.7)	27 (64.2)	32 (61.5)	2.32	0.66–8.13	0.188
9–12	19 (20.2)	11 (26.1)	8 (15.3)	3.36	0.79–14.25	0.100
**Comorbidities ^%^**	8 (8.51)	3 (7.14)	5 (9.62)	0.72	0.16–3.22	0.670

SI = secondary infection; NSI = non-secondary infection; OD = odds ratio; CI 95% = confidence interval. Association measurement not possible due to sample size. * *p* value using exact Fisher test. ^%^ In the SI, the comorbities were hypertension (*n* = 1), diabetes (*n* = 1), and rheumatism (*n* = 1); In the NSI group, the comorbities were hypertension (*n* = 4) and diabetes (*n* = 1).

## Data Availability

The data used to support the findings of this study are included within the article.

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
