# Peer review of "Inflammatory Profile Associated with Secondary Infection from Bothrops atrox Snakebites in the Brazilian Amazon"

_toxins, 2023, doi:10.3390/toxins15090524_

Round 1

Reviewer 1 Report

The article ‘Inflammatory profile associated with secondary infection from Bothrops atrox snakebites in the Brazilian Amazon’ describes an analysis of 94 patients of Bothrops atrox snake bite envenoming (SBE) in the Brazilian Amazon, highlighting the differences, on clinical and laboratory data, between patients who develop or do not develop a secondary infection (SI) to snakebite.

The data presented are interesting, what emerges is that the infection secondary to the bite appears more frequently when the site of the bite are the lower limbs (probably for a matter of hygiene), and when the effects due to poisoning are more moderate (I wonder if it is because in this case patients resort later to medical care and therefore also to the cleaning of the injured part).

Importantly, among the clinical and laboratory parameters recorded, some can be used as predictors of the development of a SI and therefore can help to provide the correct medical care.

The data are clearly presented. The introduction is appropriate, and the discussion is well argued. In the conclusions I suggest not only to mention the existence of inflammatory response profiles that are predictors of the development of a SI but to summarize which molecules can be considered SI prediction markers.

Minors:

In table 1 also specify what the abbreviations OR and IC95% stand for.

Figure 9 is not readable: the inscriptions must be enlarged.

Some sentences, I think, should be rephrased, as for example the underlined part of these:

Local effects represent a relevant clinical issue, which among manifestations inflammation signs and symptoms in the bite site represents a potential risk for short and long-term disabilities.

During follow-up, SI patients presented a worsening of local temperature, along with a sustained profile of edema and pain, while NSI patients shows a tendency to restore, and were highlighted in patients that SI was diagnosed at T2. As for laboratorial parameters, leukocytes, erythrocyte sedimentation ratio, fibrinogen and C-reactive protein were found increased in patients with SI, found more evident in patients diagnosed at T3. (What?)

The correct terms are ‘synergistic’ (not synergic) and ‘synergistically’ not synergically

Page 1 lines 10-11: ‘However, the influence of SI in the local events, still poorly understood’. The verb is missing.

Page 4 lines 109: ‘As  observed  in  Figure  2A,  both  groups  showed  a  sustained  edema  (edema  ratio above 1) from T0 to T4’. Since this is the first sentence of the paragraph, I think you should specify which are the two groups. 

Author Response

First we appreciate the efforts of Reviewer 1 on the comments, which are very important to improve the present work. All the modifications were highlighted in yellow within the manuscript.

Reviewer 1: "In the conclusions I suggest not only to mention the existence of inflammatory response profiles that are predictors of the development of a SI but to summarize which molecules can be considered SI prediction markers."

Answer: We agree with the suggestion and have summarized the markers in the conclusion

Reviewer 1: "In table 1 also specify what the abbreviations OR and IC95% stand for."

Answer: Done

Reviewer 1: "Figure 9 is not readable: the inscriptions must be enlarged."

Answer: We agree with Reviewer 1, and we have made a novel figure with enlarged inscriptions. However, although the inscriptions size improved, the overall lines of correlations in the graph have got much more confused/tangled. Therefore, we have opted to leave the graph in its original format, and would like to ask to reviewer 1 if is Ok. We understand that the figure might be displayed small within the article format, however on Toxins site format there is a possibility to enlarge the Figure and therefore its feasible to read.

Reviewer 1: "Some sentences, I think, should be rephrased, as for example the underlined part of these:"

Answer: We have revised thru all the appointed erros, and rephrased/corrected them.

Reviewer 2 Report

I have reviewed the manuscript "Inflammatory profile associated with secondary infection from Bothrops atrox snakebites in the Brazilian Amazon" and found it highly relevant. I have a few comments.

1- I suggest in the discussion to comment on the difficulty that exists in classifying an infectious process that is established in 24 hours. Most see them after 48 hours.

2- In Materials and Methods, referring to the diagnosis of secondary infection, the report of abscess or blood cultures should be included.

3- Line 30 change secondary infections to SI. Line 45 change secondary bacterial infections to secondary infections (SI). Line 56 change secondary bacterial infections to SI. Line 57 change secondary bacterial infections to SI. Line 126 change secondary infections to SI. Lead Discussion with YES. Line 342 Change erythrocyte sedimentation rate (ESR) to ESR. Line 448 change secondary infections to SI.

Author Response

First we appreciate the efforts of Reviewer 2 on the comments, which are very important to improve the present work. All the modifications were highlighted in yellow within the manuscript.

Reviewer 2: "I suggest in the discussion to comment on the difficulty that exists in classifying an infectious process that is established in 24 hours. Most see them after 48 hours."

Answer: We agree with the reviewer. As we observed in the study, 6 patients were diagnosed in 24h and 15 patients in 48h. Considering that the diagnosis of SI is based on clinical and laboratory parameters that are associated with the inflammatory process, and that envenomation also causes similar effects, we observed fewer cases of SI diagnosis at 24h due to the overlapping of the inflammatory effects of infection and snakebite envenomation. However, from 48 hours, the effects of envenomation tend to reduce, mainly due to antivenom therapy, thus making it easier to diagnosis SI based on the clinical and laboratorial results. We have added this information in the discussion section.

Reviewer 2: "In Materials and Methods, referring to the diagnosis of secondary infection, the report of abscess or blood cultures should be included."

Answer: The report of abscess (associated with cellulitis, fever, leukocytosis, lymphangitis and/or lymph- adenitisis) is already mentioned in the methods section. As concerning blood culture, this test is recommended in suspected sepsis. However, there was no cases of sepsis suspicious or confirmed. 

Reviewer 2: "Line 30 change secondary infections to SI. Line 45 change secondary bacterial infections to secondary infections (SI). Line 56 change secondary bacterial infections to SI. Line 57 change secondary bacterial infections to SI. Line 126 change secondary infections to SI. Lead Discussion with YES. Line 342 Change erythrocyte sedimentation rate (ESR) to ESR. Line 448 change secondary infections to SI."

Answer: Done